# CATS 🐱: Cross-Modal Autoencoding for Time Series Summarization

## Abstract

Time-series captioning is highly relevant in the industrial monitoring tasks: summarization of characteristic patterns and trends in time series can facilitate data analytics and enable flexible user experience. Yet, due to the scarcity of labeled data, existing data-driven methods have not seen definitive successes so far, while approaches relying on LLMs are impractical in real-world settings due to privacy, cybersecurity, and computational constraints, not to mention their big carbon footprint. In this work we ask whether a small model trained on a small dataset can produce accurate, relevant, and readable time series summaries. We propose a lightweight encoder-decoder architecture trained with a novel cross-modal autoencoding method and demonstrate that, despite its size, the model achieves performance comparable to the state-of-the-art GPT-4o and outperforms existing open-source baselines. Our results suggest that effective time series captioning is feasible under realistic industrial requirements.

## 1 Introduction

Automatic generation of time series descriptions is highly demanded in such domains as finance, weather forecasting, medicine, Internet of Things, and industry, and numerous systems were developed over the years to address this task (Kukich, 1983; Goldberg et al., 1988; Sripada et al., 2001; Ferres et al., 2006; Yu et al., 2007; Hunter et al., 2008; Carberry et al., 2013; Braun et al., 2018). Traditionally, they involved complex rule-based approaches, costly and non-generalizable. In contrast to other natural language generation (NLG) tasks (such as image captioning), data-driven time series summarization remains hampered by the scarcity of labeled datasets due to high annotation costs and confidentiality restrictions. With the recent surge in popularity of large language models (LLMs), there have been attempts to apply them to a variety of tasks, including time series classification and forecasting Gruver et al. (2024); Ansari et al. (2024). Although LLM prompting may seem appealing due to the absence of data science overhead, in practice most such approaches result in inaccurate predictions (Zhang et al., 2024), not to mention the disproportional carbon footprint of LLMs and lack of control.

Time series summarization is highly relevant in the industry, and in this work *we propose an approach to efficient and accurate time series summarization in the industrial context with limited labeled data* and develop a solution for operator decision support. To ensure safe and efficient production, plant operators monitor numerous processes. In case of anomalies or emergencies, they need to quickly identify the source of the problem and introduce corrective actions. To do so, operators could invoke a display of sensor readings or trigger a forecast of a signal. However, screen space is limited, and changing user interface in industrial control systems is a highly costly and slow process (Yaqub & Alsabban, 2023). A convenient alternative or complement to showing a plot of a signal could be its compact description. Fast automatic signal summarization would also benefit field operators, who may at best have a mobile device with a small screen or none at all.

A text or audio message, e. g. *'Temperature in boiler X was stable and then increased sharply over the past several minutes'*, instead of [00:00:00 52.4676; 00:01:00 51.5642; 00:02:00 52.8840; 00:03:00 53.6723; 00:04:00 52.3768; 00:05:00 54.6674; 00:06:00 57.7941; 00:07:00 60.7138; 00:08:00 64.8411; 00:09:00 69.7367; 00:10:00 73.6085; 00:11:00 76.4305; 00:12:00 77.3991; 00:13:00 79.7648; 00:14:00 80.8566], could be highly helpful and, unlike a plot, easy to integrate. In the industry, safety is paramount, therefore, a quick response of the operator in case of failure is critical to prevent losses, downtime, and hazards to the environment and human health. For this reason, prediction speed and accuracy are crucial. In addition, due to stringent cybersecurity and reliability requirements, many industrial DCS and SCADA systems are air-gapped from external networks, therefore, models must run offline (Knapp, 2024). They also need to be fast and sufficiently compact to be deployable across various control systems, most of which are CPU-only.

In view of the above, there are several *requirements to time series summarization*: *faithfulness to input* (truthfully describing relevant properties of time series), *readability* (grammatical and stylistic correctness). Practically, *prediction speed*, as well as *independent and efficient offline deployment* preserving *privacy* of customer data, are critical factors for industrial applications.

To satisfy these requirements, we propose Cross-modal Autoencoding for Time series Summarization (CATS) for accurate and efficient generation of time series descriptions using a model trained with a novel technique of cross-modal autoencoding on a small dataset. Specifically, we make the following contributions: (1) a compact multimodal model, CATS, consisting of a time series encoder and a text decoder, which can be efficiently pretrained on unlabeled data; (2) a novel cross-modal autoencoding method as a training technique for multimodal NLG with limited annotated data; (3) a new metric, TrendScore, providing a realistic evaluation of time series descriptions, in contrast to traditional NLG scores, which we show to be misleading for this task; (4) a demonstration of a successful application of our solution on describing real industrial data.

## 2 Related Work

For a long time, time series captioning was on the margin of NLG research. Existing works involved elaborate rule-based systems relying heavily on linguistic knowledge and domain heuristics (Kukich, 1983; Goldberg et al., 1988; Sripada et al., 2001; Yu et al., 2007; Portet et al., 2009; Carberry et al., 2013; Braun et al., 2018). These systems were highly costly and non-generalizable (Reiter & Dale, 1997; Gatt & Krahmer, 2018).

Even after the advent of rather capable language models (LMs), surprisingly few studies tried using them in the time series domain and building data-driven systems. Murakami et al. (2017) generate stock market comments with an encoder-decoder model trained on a dataset of 7351 samples. Using embeddings of long- and short-term time series, a decoder generates placeholders replaced at test time with arithmetic operations to copy numeric values. Jhamtani & Berg-Kirkpatrick (2021) first learn to identify a small predefined set of patterns in time series, which a used by an LSTM to generate a description. 5700 stock price samples were crowdsourced for the task. Harris & Zaki (2022) generate simple summaries of health data with an encoder-decoder model learning a template and identifying values to fill in. Data from 9.9k users was annotated with rule-based methods, which constrained generation capabilities. Cai et al. (2024) pre-train a model on 14,650 samples on reconstruction and matching tasks before fine-tuning on captioning. Only limited examples are provided.

In view of the surge in popularity of LLMs, a number of studies attempted applying them to time series. An overview of such approaches is provided by Zhang et al. (2024), namely: prompting (passing time series to LLMs directly as raw text (Xue & Salim, 2023; Gruver et al., 2024)), quantization (discretizing time series into bins (Ansari et al., 2024)), aligning (learning time series embeddings aligned with language (Jin et al., 2023)), vision as bridge (plotting time series and using vision-language models (Zhang et al., 2023)), tool integration (adopting LLMs to output dedicated tools). Most of the studies deal with tasks such as forecasting or classification and achieve performance that is overall on par with existing models (which are usually much more compact and efficient). Few studies tackle time series description, such as Zhang et al. (2023), yet no conclusive evaluation is available. In more recent works, time series representations are aligned with LLM embeddings:

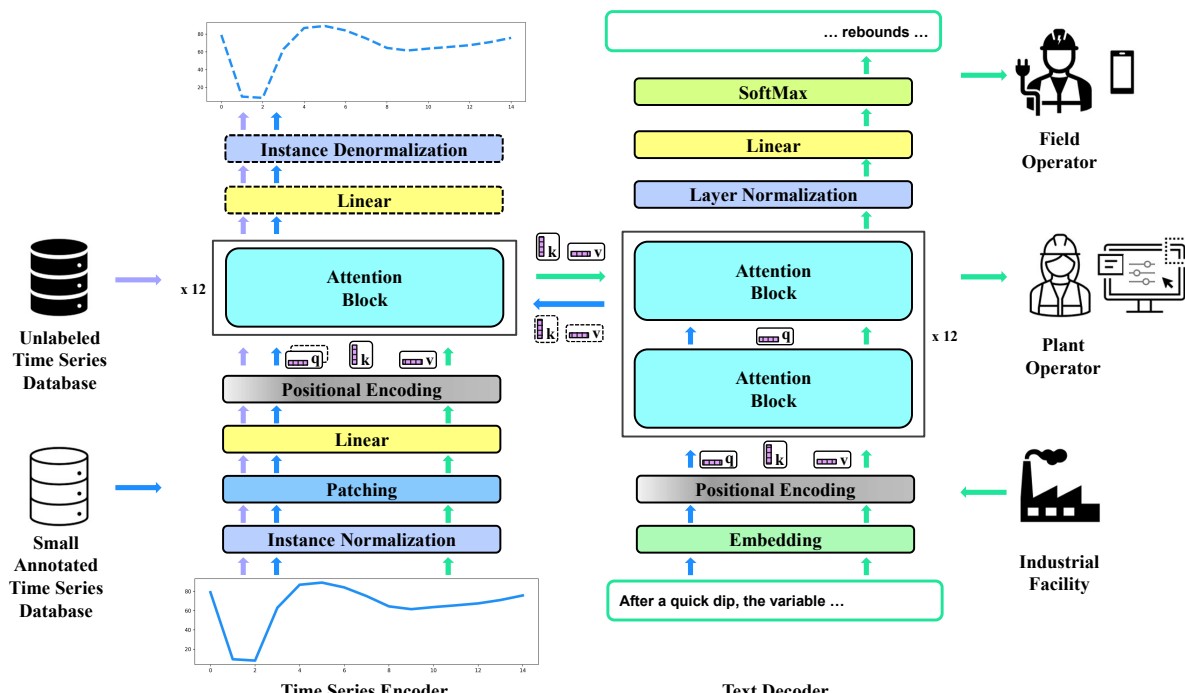

Figure 1: CATS architecture: during the main training stage, text is generated based on a time series embedding and preceding text tokens (green arrows). The flow is the same at inference, when signal is received from the plant and the generated summary is presented to the operator. During the cross-modal autoencoding pretraining stage, text embeddings are fed back into the encoder to recreate the time series (blue arrows). Optionally, the model can be pretrained with unlabeled data (lilac arrows)

Chow et al. (2024) concatenate outputs of a time series encoder with text embeddings and feed them to an LLM; the encoder is trained while prompting a frozen LLM with classification and captioning tasks; Trabelsi et al. (2025) first train a model to align time series as string tokens with text embeddings, and then generate several candidates for input time series and prompt an LLM to summarize them.

# 3 CATS: Generating Time Series Descriptions

To enable efficient generation of time series descriptions that are not only grammatical and fluent but also faithful to input and its most characteristic properties, we need a model that is compact, supports transfer learning, and can be trained to capture a mapping between patterns in the source modality (time series) and references to them in the target modality (text), even with little data, which is always scarce in the industry. The proposed architecture and training technique that fulfill these properties are described below.

## 3.1 Model Architecture

For efficient summarization, the model is composed of a dedicated time series encoder, PatchTST (Nie et al., 2022), and a lean text decoder, GPT-2 (Radford et al., 2019), connected via cross-attention (Vaswani et al., 2017), which has been successfully adopted in numerous works (Ondeng et al., 2023). A pretrained GPT-2 generates fluent text while being compact, and PatchTST has shown robust performance across a variety of look-back windows and prediction horizons and can be configured to match GPT-2 in the number of layers (12), attention heads (12), and the embedding dimension (768). It can also be pretrained with unlabeled data.

The architecture is shown in Fig. 1. In the encoder, since time series are not stationary and come from different sensors, each window is independently standardized, then divided into patches of length 5 with stride 3. The patches go through a linear layer, and sine-cosine positional encoding is added to its output, which is then fed into the attention block. Each such block includes self-attention (with query, key and value coming from the time series), a feed-forward layer and a residual connection. The output is passed to the decoder cross-attention block (with key-value pairs from the time series, and query from its caption).

Descriptions are generated autoregressively, beginning from the 'BOS' token. Tokens are passed into the decoder embedding layer, then positional encoding is added to the embeddings, which are further passed to the decoder self-attention blocks (with query, key and value all coming from the text modality) and cross-attention blocks (with query coming from the text and key and value pairs — from the encoder self-attention block). The self-attention block in the decoder includes causal masking, which prevents the model from attending to future tokens. The output of the final attention block is passed to a linear layer, followed by softmax, producing next-token probabilities for text generation. Thus, during training on labeled data, the model learns the conditional probability of a sequence of text tokens $Y$ of length $N$ given time series $X$: $P(Y \mid X) = \prod_{i=1}^{N} P(y_i \mid y_1, \ldots, y_{i-1}, X)$. Self-attention is first applied to the target sequence, capturing the probability of each token given the past tokens, and subsequently, cross-attention ensures that the text is learned in alignment with the time series data. During training, categorical cross-entropy, $-\frac{1}{N} \sum_{i=1}^{N} \sum_{k=1}^{V} y_{i,k} \log(\hat{y}_{i,k})$, where $V$ is vocabulary size, is minimized over the decoder output.

### 3.2 Cross-Modal Autoencoding

Although cross-attention enables the flow of information from the time series encoder to the text decoder, *the model may still overfit to the text distribution and the time series vocabulary without necessarily learning to map words back to patterns in time series.* Categorical cross-entropy would not be sufficient to control that, since a crucial word, such as 'increase', 'oscillate', or 'plummet', is but one token in a sequence, compare: *'The speed decreases slightly and stabilizes for a short time, then rises again...'* with a candidate description *'The speed increases slightly and stabilizes for a short time, then drops again...'* Only 2 of 13 words are wrong, and the loss would be low, yet these two words completely flip the meaning. See also Sec. 4.3.2.

To prevent such shortcuts, we propose to strengthen the alignment of two modalities through a novel pretraining method: cross-modal autoencoding. Intuitively, the trends of a time series should be reconstructible from its description (relying on words such as 'climb', 'fall' or 'stabilize'). Using this intuition, we pretrain on a reverse task: recreating time series from text. The model thus learns the probability of a time series $X$ of length $M$ given word tokens $Y$: $P(X \mid Y) = \prod_{j=1}^{M} P(x_j \mid x_1, \ldots, x_{j-1}, Y)$. In this case, embeddings of the description are fed back into the encoder in cross-attention mode (with key, value pairs coming from the text, and query — from the self-attention layer of the encoder). This flow is marked by blue arrows in Fig. 1. Mean Squared Error (MSE) between the recreated and original time series, $\frac{1}{M} \sum_{j=1}^{M} (x_j - \hat{x}_j)^2$, is back-propagated, reinforcing the learned mapping between time series patterns and phrases in their descriptions.

### 3.3 Multimodal Training

To generate accurate, relevant and readable time series summaries, the model must learn salient representations of both modalities and a robust mapping between them. To this end, several strategies are used.

To learn robust representations of time series, the **encoder** is first pretrained on the classical tasks of forecasting and autoencoding in a in a self-supervised manner, utilizing unlabeled data. In forecasting, at time $t_0$, based on a window of $N$ minutes from $t_{0-N}$ to $t_0$, the model is to predict the next data point $t_1$. The autoencoding task involves reconstructing a time series window from $t_0$ to $t_N$. Pretraining is done by minimizing the MSE loss. In our experiments, we compare model performance with and without pretraining.

The **text decoder** is initialized with open-source GPT-2 weights to foster fluency and grammatical correctness of generated text. At inference, a temperature of 0.5 was used to control randomness in token sampling.

The **model** as a whole is trained on the main task of time series description: for a time series from $t_0$ to $t_N$, it generates a caption, which is compared to the ground truth, and categorical cross-entropy is minimized.

For stronger multimodal representation learning, the model is pretrained using the **novel cross-modal autoencoding method**, whereby the model first generates textual descriptions of time series and then reconstructs the time series based on the produced text, simultaneously minimizing the loss on both tasks (categorical cross-entropy for text and MSE for time series). For balanced training, the text generation loss and the time series reconstruction loss are scaled using a coefficient $\alpha$ (empirically chosen as 0.5) such that the combined loss for time series $X$ of length $N$ and its textual description $Y$ of length $M$ with vocabulary size $V$ is:

$$L = \alpha - \frac{1}{N} \sum_{i=1}^{N} \sum_{k=1}^{V} y_{i,k} \log(\hat{y}_{i,k}) + (1 - \alpha) \sum_{j=1}^{M} (x_j - \hat{x}_j)^2 \tag{1}$$

To enable reconstruction, the model is first trained to minimize the cross-entropy loss on text generation, and once it drops below 1, the decoder is frozen, and the cross-modal loss is minimized. To avoid overfitting, after 5 epochs, all layers are unfrozen and normal training is resumed, minimizing cross-entropy. This method allows training an accurate text generation model, faithfully describing time series, without expensive data annotation — by relying on a small labeled dataset.

Training was done using Adam optimizer with a learning rate of $4 \times 10^{-6}$ and early stopping (with a patience of 5 epochs and min. delta 0.001) for 25-30 epochs. All experiments were run with 10 random seeds.

## 4 Experimental Setup

Below we describe the collected data as well as the baselines and the evaluation methodology.

### 4.1 Data Collection

One important problem hindering publications on the topic of automatic time series summarization is *data scarcity* and *absence of benchmarks*. The few existing studies (see Sec. 2) annotate their own data (occasionally using templates), which often appears too simplistic. Moreover, existing work focuses on specific domains, such as weather or medicine, with characteristic lexical and stylistic traits, which limit their generalizability (consider words like 'cloudy', 'feverish', or 'skyrocketing stock prices'). Of the few studies, only one published the data. It comes from the financial domain, which reflects in the text style, but descriptions are sufficiently generic for a quantitative test. The evaluation on this dataset is reported in Sec. 5.4.

Due to the limitations of this dataset (see details in Sec. 5.4), we collected our own data both for training and qualitative testing. We used a proprietary industrial dataset that included recorded temperatures, pressures, flows, and other signals. To model real-world use cases, we randomly sampled 10-minute windows (a length common in forecasting and similar tasks in production), which were annotated by 41 Master's students, each describing at least 10 samples. Since this work is driven by industrial requirements, the annotators were instructed to describe samples accurately, yet concisely, capturing the main patterns and trends, and using the word 'Variable' as a placeholder, such that in production it could be substituted with 'temperature', 'speed', 'level', etc. Descriptions had to be stylistically neutral and generic. The resulting dataset consisted of 416 samples (split into 250 train, 66 dev and 100 test). In addition, 6162 unlabeled time series windows were randomly sampled for pretraining. To compensate for the modest dataset size, we carried out an extensive human evaluation: the 100 test samples were each rated by 5 users, resulting in 500 evaluated time series and 1320 description candidates in total — with moderate intraclass correlation (Shrout & Fleiss, 1979): ICC3 = 0.337, p-value 0.05. Importantly, absence of good-quality open datasets of time series descriptions motivates this work, and the contributed novel cross-modal autoencoding method is aimed to train sufficiently accurate models even with little data.

### 4.2 Baselines

Time series summarization is a very young field, with *no established baselines*. At the same time, given the current surge of interest to LLMs, a number of studies applied LLMs such as GPT-2 and Llama 2 to time series (see Sec. 2). Following these, we used several LLMs as a baseline, prompted to describe time series in three different formats: as *raw values* (floating-point time series are passed as a

Table 1: Automated metrics: classical scores do not reflect the accuracy of trend descriptions: the semantically similar examples 1-4 are scored lower than 5-7, where predictions are wrong

| Prediction | Reference | BLEU | ROUGE | SPICE | METEOR |
|---|---|---|---|---|---|
| 1 Value declines gradually | Variable experiences a decreasing trend | 0.00 | 0.00 | 0.00 | 0.00 |
| 2 Variable climbs up | Variable experiences an increasing trend | 0.17 | 0.25 | 0.33 | 0.10 |
| 3 Variable has a stationary trend | Variable maintains the same stable value | 0.13 | 0.17 | 0.22 | 0.14 |
| 4 Variable experiences fluctuations | Variable oscillates between a higher and a lower values | 0.00 | 0.07 | 0.00 | 0.03 |
| 5 Variable maintains a constant value followed by an increasing trend in value | Variable demonstrates an overall decreasing trend | 0.23 | 0.32 | 0.31 | 0.16 |
| 6 Variable climbs up the incline and is stable at the end | Variable oscillates between values and has a lot of noise | 0.27 | 0.27 | 0.18 | 0.12 |
| 7 Variable demonstrates an overall decreasing trend | Variable experiences an increasing trend | 0.50 | 0.55 | 0.44 | 0.21 |

string directly); as *rounded values* (time series are rounded off to integers and passed to an LLM as a string); as *Symbolic Aggregate Approximation (SAX)* (data is binned into a sequence of discrete values). All models including CATS were prompted with: 'Describe the pattern or trend of the following time series in one sentence capturing its main properties: avoid references to exact times and values. Use the term 'variable' to refer to the time series'. Cf. Trabelsi et al. (2025): 'Describe this time series <time series> encoded by <time series embedding>'.

Our research is driven by industry demand. As we briefly explain in Sec. 1, industries have strict *data privacy* and *cybersecurity* requirements, and control systems commonly operate offline, often on CPU-only legacy hardware, which limits the choice of models to open-source LLMs deployed locally (similar limitations also apply to other domains, e.g. healthcare). Other critical factors are *prediction speed* and *computational resources*, therefore, the models need to be compact, so even 7B-scale models typically cannot be deployed in practice. In view of these criteria, we used several legally approved locally available LLMs as a baseline: GPT-2, instruction-tuned Mistral 7B (v0.3) (Jiang et al., 2023), Llama 2 (Touvron et al., 2023), Llama 3.1 and Gemma 7B (Team et al., 2024), and GPT-4o on the public financial dataset. For the sake of completeness, in initial experiments, we also tested vision-language models, but abandoned the path due to poor results — cf. Llava 7B summary of Fig. 4 (a): *The image shows a downward trending line graph representing volume fluctuations, with two lines indicating different sets of data.* Overall, since 1D time series are much more compact than a 3D plot, we did not consider applying image captioning, which addresses different challenges, but rather focused on the underexplored problem of time series summarization.

### 4.3 Evaluation Methodology

We now discuss the limitations of traditional NLG metrics in the context of time series description and explain the proposed alternative.

#### 4.3.1 Automated Metrics

Traditional NLG metrics such as BLEU (Papineni et al., 2002) or ROUGE (Lin, 2004), although used to evaluate time series summarization (see Sec. 2), are ill-suited for the task, since they rely on n-gram overlap and ignore synonyms and word order: in Tab. 1, examples 1-4 convey the same meaning using synonyms yet receive lower scores than examples 5-7, where predictions are incorrect. The same concerns embedding-based metrics such as BertScore (Zhang et al., 2019), which measures similarity of contextualized BERT embeddings of source and target tokens. By visualizing BERT embeddings (see Fig. 2), one can see that, e.g., words like declined, increase, and stable are close together but far from their respective synonyms. Such inconsistency

in embeddings of words that are critical for semantic accuracy of time series descriptions invalidates metrics that rely on such embeddings for their evaluation. In addition, even assuming a meaningful mapping of words describing time series trends, the score would be biased towards the overall similarity of a token sequence, which could be high even if crucial words are wrong, and some kind of weighing and tuning would be necessary to ensure that a metric focuses on relevant words.

### 4.3.2 TrendScore

Given that popular text similarity metrics are unreliable for the task, we propose a new metric: TrendScore, specifically designed to assess time series descriptions. In this approach, descriptions are categorized into *six trend classes: increasing; decreasing; stable; increasing and then decreasing; decreasing and then increasing; and fluctuating.* The classes were determined by inspecting the annotated data. We collected an extensive list of keywords corresponding to each class (148 in total). After POS tagging and lemmatization, based on keyword presence, order and combination with location words, each description is assigned a trend class - see pseudocode in Listing 1 (e.g. keywords and combinations *rebounds; troughs and ascends; lowpoint in the middle* all indicate the class *decreasing and then increasing*). When all reference and predicted sentences are classified, F1 is calculated .

This way, captions describing different trends are considered different classes even if descriptions were structured similarly or differed in only one word. Although this method has limitations (neither the set of trend classes nor keywords can be exhaustive), nonetheless it serves as a much more appropriate and rigorous evaluation method of time series descriptions than classical NLG metrics. For the sake of this work, we manually verified the classes assigned to test set samples. In addition, to avoid any bias, we conducted a user survey for a more nuanced evaluation of our model.

### 4.3.3 Human Evaluation

To ensure a holistic and impartial evaluation, we conducted an extensive user study. We split 100 test samples into 9 forms, each rated by five users: 22 Master's students and 23 researchers and engineers (resulting in

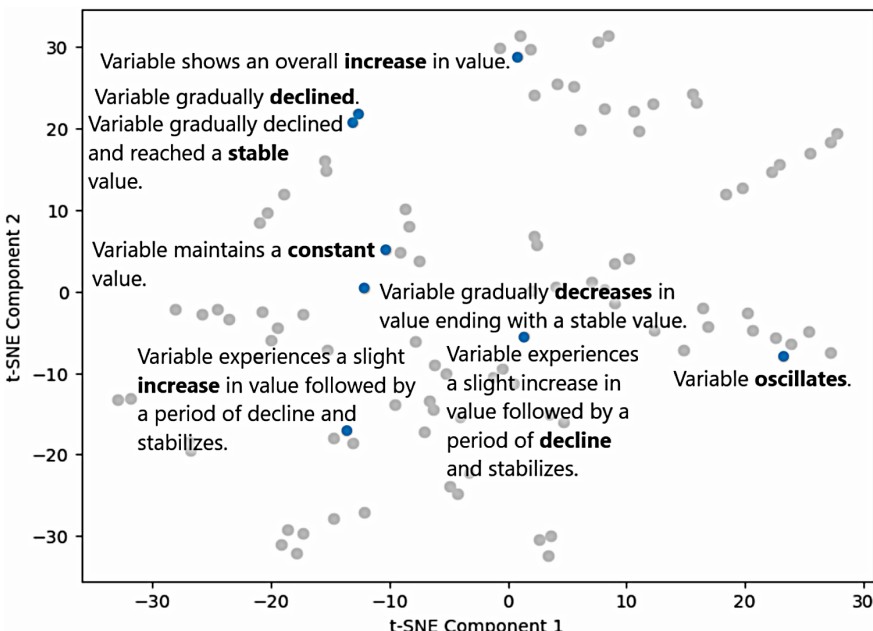

Figure 2: t-SNE visualization of BERT embeddings: trend words (**bold**) are plotted in blue, others in gray. Embeddings do not reflect the meaning of trend words, e.g. declined is closer to its antonym increase than to synonyms decreases and decline

Listing 1: Sentence classification pseudocode

```python
def classify_sentence(lemmatized_sentence):
    increase_words = {'increasing', 'rise', 'grow' ...}
    decrease_words = {'decreasing', 'fall', 'drop' ...}
    oscillate_words = {'oscillate', 'fluctuate', 'noise' ...}
    stable_words = {'stable', 'steady', 'consistent' ...}
    max_words = {'peak', 'top', 'apex' ...}
    min_words = {'dip', 'lowpoint', 'trough' ...}
    bounce_words = {'rebound', 'bounce', 'recover' ...}
    start_words = {'start', 'begin', 'initial' ...}
    end_words = {'end', 'final', 'last' ...}
    middle_words = {'middle', 'midpoint', 'midway' ...}

    if any(word in tokens_lemmatized for word in increase_words):
        if any(word in tokens_lemmatized for word in decrease_words):
            for inc_word in increase_words:
                for dec_word in decrease_words:
                    if sentence.find(inc_word) < sentence.find(dec_word):
                        return 'increasing_and_then_decreasing'
                    elif sentence.find(dec_word) < sentence.find(inc_word):
                        return 'decreasing_and_then_increasing'
    if any(x in increase_words for x in tokens_lemmatized) and not any(
        x in decrease_words for x in tokens_lemmatized):
        return 'increasing'
    elif any(x in decrease_words for x in tokens_lemmatized) and not any(
        x in increase_words for x in tokens_lemmatized):
        return 'decreasing'
    elif any(x in oscillate_words for x in tokens_lemmatized):
        return 'oscillating'
    elif any(x in stable_words for x in tokens_lemmatized):
        return 'stable'
    elif any(x in max_words for x in tokens_lemmatized):
        if any(x in start_words for x in tokens_lemmatized):
            return 'decreasing'
        elif any(x in end_words for x in tokens_lemmatized):
            return 'increasing'
        elif any(x in middle_words for x in tokens_lemmatized):
            return 'increasing_and_then_decreasing'
    elif any(x in min_words for x in tokens_lemmatized):
        if any(x in start_words for x in tokens_lemmatized):
            return 'increasing'
        elif any(x in end_words for x in tokens_lemmatized):
            return 'decreasing'
        elif any(x in middle_words for x in tokens_lemmatized):
            return 'decreasing_and_then_increasing'
    elif any(x in bounce_words for x in tokens_lemmatized):
            return 'decreasing_and_then_increasing'
    else:
        return None
```

500 time series and 1320 description candidates in total). *First*, users were asked to rate descriptions of 4 time series based on three criteria on a Likert scale from 1 to 7: **relevance** (how well a summary captures the most important aspects of the time series), **accuracy** (how truthfully a caption represents the direction, steepness, and sequence of trends), and **readability** (how easy the text is to read and understand). *Second*, users were asked to **rank several captions** for 7 time series based on how well they described the input: 3 samples from the test set (including a human description) and 4 unlabeled samples.

## 5 Results and Discussion

Below we discuss the experimental results. We compare CATS to the baselines (see Sec. 4.2) and run an ablation with other pretraining strategies (see Sec. 5.2): we compare models with the same architecture as ours, however, our model, with time series encoder pretrained with the novel cross-modal autoencoding (*CATS, CM\**), is compared to models with encoder pretrained on classical autoencoding (*CATS, AE*), forecasting (*CATS, FO*) and no pretraining (*CATS, NP*) — all trained on the time series description task.

### 5.1 Accuracy and Efficiency

The overall statistics and the TrendScore as a measure of accuracy of time series descriptions (see Section 4.3.2) are summarized in Tab. 2. Compared to the baselines (see Section 4.2), CATS shows several obvious advantages. Most importantly, it achieves a TrendScore of *0.91*, which is significantly higher than an average LLM. Although there is some variation both among the LLMs and across time series encoding types, their overall score is unacceptably low, clearly showing that a compact

Table 2: Comparing CATS to the baselines: in addition to an incomparably superior accuracy in terms of TrendScore (as well as BertScore and LLM-as-a-judge), CATS\* is much more compact and quick at inference (at a negligible training overhead, compared to the massive training done by the LLM providers)

| Model | Input | Size (B) | Train Time (s) | Inference Time (s) | Trend Score | Bert Score | llama3.1 F1 | llama3.1 Consist. |
|---|---|---|---|---|---|---|---|---|
| | | | **CATS\* with Pretraining Variants** | | | | | |
| CATS, NP | time series | 0.24 | 179.0 | 0.2 | 0.75±0.04 | 0.93±0.01 | 0.70±0.05 | 0.73±0.02 |
| CATS, FO | time series | 0.24 | 145.4 | 0.3 | 0.78±0.04 | 0.93±0.01 | 0.61±0.05 | 0.76±0.04 |
| CATS, AE | time series | 0.24 | 185.3 | 0.2 | 0.83±0.03 | 0.94±0.01 | 0.66±0.05 | 0.78±0.03 |
| CATS, CM\* | time series | 0.24 | 230.0 | 0.3 | 0.91±0.04 | 0.94±0.01 | 0.71±0.06 | 0.76±0.05 |
| | | | **Prompting Off-the-Shelf Models** | | | | | |
| GPT-2 | raw string | 0.12 | - | 0.8 | 0.06±0.02 | 0.78±0.0 | 0.10±0.05 | 0.65±0.03 |
| GPT-2 | rounded | 0.12 | - | 1.0 | 0.06±0.01 | 0.77±0.0 | 0.07±0.03 | 0.56±0.02 |
| GPT-2 | sax | 0.12 | - | 1.1 | 0.21±0.01 | 0.77±0.0 | 0.08±0.04 | 0.65±0.03 |
| Llama2 | raw string | 6.74 | - | 18.9 | 0.30±0.05 | 0.85±0.0 | 0.33±0.05 | 0.69±0.03 |
| Llama2 | rounded | 6.74 | - | 17.4 | 0.29±0.03 | 0.85±0.0 | 0.29±0.03 | 0.73±0.03 |
| Llama2 | sax | 6.74 | - | 18.1 | 0.21±0.02 | 0.85±0.0 | 0.11±0.05 | 0.82±0.03 |
| Llama3.1 | raw string | 8.03 | - | 23.1 | 0.44±0.04 | 0.86±0.0 | 0.32±0.04 | 0.73±0.04 |
| Llama3.1 | rounded | 8.03 | - | 19.7 | 0.43±0.02 | 0.86±0.0 | 0.26±0.05 | 0.78±0.03 |
| Llama3.1 | sax | 8.03 | - | 20.5 | 0.27±0.02 | 0.86±0.0 | 0.22±0.06 | 0.78±0.05 |
| Mistral | raw string | 7.25 | - | 70.0 | 0.48±0.04 | 0.86±0.0 | 0.18±0.03 | 0.80±0.03 |
| Mistral | rounded | 7.25 | - | 79.2 | 0.44±0.05 | 0.86±0.0 | 0.16±0.04 | 0.82±0.03 |
| Mistral | sax | 7.25 | - | 86.9 | 0.32±0.01 | 0.84±0.0 | 0.12±0.03 | 0.86±0.03 |
| Gemma | raw string | 8.54 | - | 15.4 | 0.41±0.04 | 0.86±0.0 | 0.28±0.06 | 0.74±0.05 |
| Gemma | rounded | 8.54 | - | 14.3 | 0.38±0.08 | 0.86±0.0 | 0.21±0.04 | 0.80±0.02 |
| Gemma | sax | 8.54 | - | 16.9 | 0.35±0.04 | 0.85±0.0 | 0.19±0.05 | 0.79±0.05 |

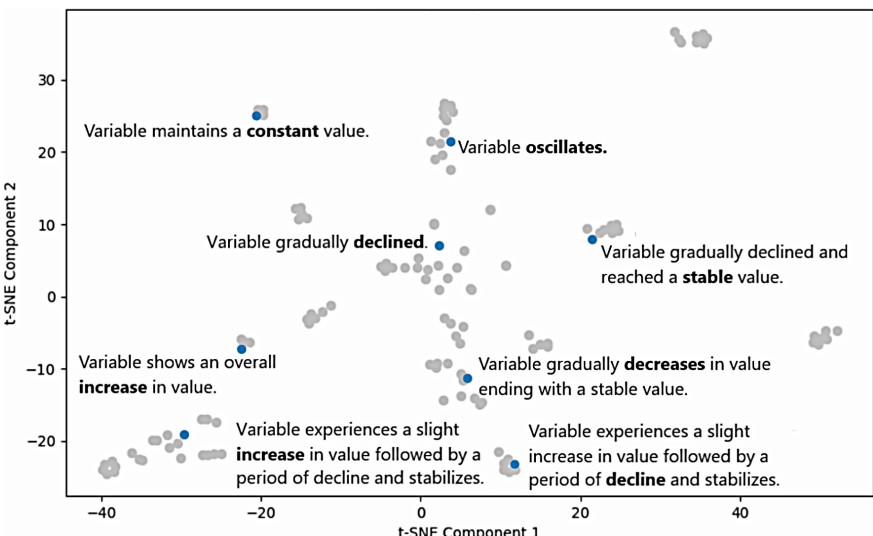

Figure 3: t-SNE visualization of embeddings from CATS pretrained with cross-modal autoencoding: trend words (**bold**) are plotted in blue, others in gray. Embeddings are aligned with the meaning of trend words, e.g. declined, decreases and decline are well separated from the two instances of increase

dedicated model trained even on a small dataset is much better suited for generating time series descriptions than an LLM without specialized training (at least at the time of the experimental work). In contrast to TrendScore, BertScore is indiscriminately high for all models, regardless of the quality of summaries (see examples in Fig.4: descriptions output by LLMs are typically very generic and not sample-specific). We argue that this only reflects the fact that texts belong to the same topic — time series. This also explains the bias of BertScore towards CATS: the higher scores are likely due to the ground truth annotations in the test set coming from the same distribution as the train set (see also Sec.5.4).

Additionally, we implemented LLM-as-a-judge with Llama3.1: for each sample, the model was prompted to evaluate how well the predicted summary captures the overall pattern or trend of the time series compared to the reference description. Based on the output, binary F1 was calculated. While the resulting scores for CATS variants ranged between 0.61 and 0.71, all LLMs (including Llama3.1 itself) received F1 between 0.07 and 0.33. Nonetheless, without a proven reliability of the LLM for the task, its judgments should be interpreted with caution: first, descriptions produced by Llama3.1 are too generic (see Fig.4) and there is no reason to expect that it can evaluate summaries better than generate them; second, prompted three times for each sample, for most models it returned the same result only 60-70% of the time.

Importantly, the considerably smaller model size (ca. $32 \times smaller$ than 7B-scale LLMs), faster inference (ca. *4 to 400 ×faster*), and fast and cheap training (compared to the massive training on the side of LLM providers) make our solution an efficient and environment-friendly alternative to LLMs, especially considering the requirements of *reproducibility*, *confidentiality*, and *independent offline deployment*, which are imperative in the context of commercial and industrial applications.

## 5.2 Impact of Pretraining

We have demonstrated that a compact model trained for time series summarization by far outperforms huge general-purpose LLMs. To assess the effect of the novel cross-modal autoencoding method, we ran an ablation study comparing four instances of CATS with identical architecture and training, but different pretraining.

With respect to faithfulness of descriptions to the input, measured with **TrendScore** (see Tab. 2), unsurprisingly, pretraining the time series encoder on autoencoding and forecasting (whereby it learns temporal dependencies and trends in the input) benefits the subsequent time series captioning, increasing TrendScore

by 4.00% and 10.66%, respectively, compared to no pretraining. Yet cross-modal autoencoding offers a clear advantage over the other methods, which is even more prominent compared to no pretraining (by **21.33%**).

In terms of **user ranking**, one can observe that descriptions written by humans are preferred over any model-generated summaries in 0.67% of the cases (see Tab. 3). However, there is no consistent leader among the pretraining strategies: e.g. the model with autoencoding is chosen best least often, but most often — second best, and differences are negligible. We hypothesize that this is due to fatigue and attention span limit: after choosing the best variant (and the choice is often non-obvious), one would be less attentive and motivated to rigorously compare and order other candidates. A clearer distinction is revealed when human annotations were not available (see Tab. 4). The model pretrained with cross-modal autoencoding is ranked both best and second best most often, followed by unimodal autoencoding, confirming the TrendScore and showing that time series encoder pretraining helps learn better representations and generate more accurate descriptions. Compared to the classical techniques, cross-modal autoencoding demonstrates even more robust results.

The superiority of cross-modal autoencoding also manifests itself **visually**. Fig. 3 shows CATS embeddings of trend words. One can see clear grouping of words increase; declined, decreases and decline; constant and stable, and oscillates. The last three look like one group, which also makes sense, as oscillated time series can be viewed as stable but noisy signal, as opposed to an upward or downward trend. Fig. 3 is a stark contrast to Fig. 2. The clear separation between the embeddings of words referring to increasing and decreasing trends from those related to constant and oscillating signals indicates that due to cross-modal autoencoding, CATS, CM* learned to successfully distinguish different time series patterns.

### 5.3 Parametrized Evaluation

For a more nuanced evaluation, we asked users to assess descriptions by CATS, CM* according to the criteria of relevance, accuracy, and readability on a Likert scale from 1 to 7. **Relevance** and **accuracy** reached a rating of *5.58±1.50* and *5.33±1.60*, respectively. The similar scores on both criteria make sense, since a description that is relevant (capturing the most important aspects of a time series window) must also be accurate (truthfully reflecting the direction, steepness, and sequence of trends). The high scores testify that, on average, summaries captured the main characteristics of time series and described them faithfully. **Readability** scored even higher, with an average of *6.34±1.07*, suggesting that summaries generated by CATS, CM* were generally grammatically correct and easy to read and understand. The intra-class correlation (Shrout & Fleiss, 1979) was moderate (ICC3=0.34, p-value=0.05).

Table 3: User ranking of summaries by humans and by CATS with different pretraining in the order of preference (human version predictably preferred)

| Source | $1^{st}$ choice | $2^{nd}$ choice | $3^{rd}$ choice | $4^{th}$ choice |
|---|---|---|---|---|
| Human | 0.67 | 0.16 | 0.11 | 0.06 |
| CATS, NP | 0.15 | 0.28 | 0.28 | 0.29 |
| CATS, AE | 0.07 | 0.35 | 0.41 | 0.17 |
| CATS, CM* | 0.11 | 0.22 | 0.19 | 0.48 |

Table 4: User ranking of summaries by CATS with different pretraining in the order of preference: cross-modal autoencoding is most frequently chosen as both first and second best

| Source | $1^{st}$ choice | $2^{nd}$ choice | $3^{rd}$ choice |
|---|---|---|---|
| CATS, NP | 0.25 | 0.27 | 0.48 |
| CATS, AE | 0.36 | 0.30 | 0.34 |
| CATS, CM* | 0.39 | 0.43 | 0.18 |

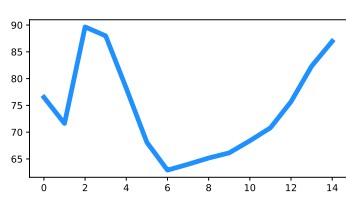

(a) **Human:** In the beginning the variable stays on the same value until it grows rapidly for a short time before falling rapidly before growing slowly again.
**CATS:** Variable is stable on the same value until it starts growing after reaching its highest point falls fast a short time to grow at the end.
**Mistral:** Variable exhibits fluctuations around two distinct levels, with periods of relatively low values clustered around 65 and periods of higher values around 85.
**Gemma:** The time series exhibits an alternating pattern of increasing and decreasing values, with a general trend of gradual decline.
**Llama3.1** : The variable exhibits fluctuations around a general trend with periods of increase and decrease that can be observed over time.

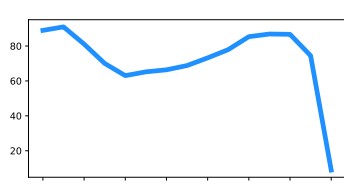

(b) **Human:** Variable decreases and grows slightly and at the end decreases rapidly.
**CATS:** Variable after rising, slowly decreases and then gradually increases again but in the end, shows a sharp drop.
**Mistral:** Variable exhibits a fluctuating trend with some values increasing significantly while others decrease noticeably.
**Gemma:** The time series exhibits an oscillating pattern with a general downward trend, punctuated by occasional spikes and rebounds.
**Llama3.1** : The variable exhibits a general trend of fluctuating values over time, with some periods of increase and decrease in magnitude.

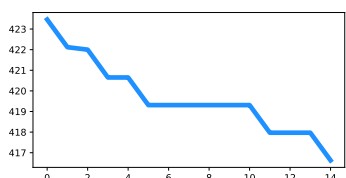

(c) **Human:** Variable declines in a stable manner.
**CATS:** Variable demonstrates an overall decreasing trend.
**Mistral:** Variable exhibits a flat or stationary trend with minor fluctuations around specific values towards the end.
**Gemma:** The time series exhibits a pattern of alternating fluctuations between slightly higher and lower values, with a general trend of decreasing values over time.
**Llama3.1** : The variable generally exhibits a gradual decline over time, with periods of stability or brief reversals in the trend before continuing downward.

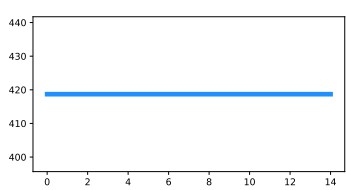

(d) **Human:** Variable maintains the same stable value.
**CATS:** Variable displays no change in value.
**Mistral:** Variable exhibits a consistent and stable value with no discernible trend or variation over the observed time period.
**Gemma:** Variable exhibits a repetitive pattern of being constant at 418.70 for each element in the sequence.
**Llama3.1** : The variable exhibits a consistent and unchanging pattern over the observed period, with no discernible trend or fluctuation in its value.

Figure 4: Examples of time series descriptions. In contrast to non-specific LLMs' descriptions, occasionally hallucinating absent properties, in most cases CATS captures trends and patterns in time series faithfully

A few examples are shown in Fig. 4. One can make several observations. First, the annotators were not native speakers; therefore, their descriptions were not without imperfections, which also propagated to the model. However, LLMs do not show a consistent advantage in readability or fluency and tend to be unnecessarily verbose. More importantly, most descriptions produced by LLMs are very vague and fail to capture the observed patterns, occasionally hallucinating absent properties. By contrast, in most cases CATS faithfully captures the patterns in time series. To sum up, although there is still room for improvement, the time series descriptions generated by our model were generally reliable, accurate and readable.

## 5.4 Evaluation on a Financial Dataset

Datasets of time series paired with descriptions are extremely scarce, however, there exists a stock price dataset, TRUCE (Jhamtani & Berg-Kirkpatrick, 2021), containing 1900 real and 560 simulated time series samples, each described by 3 annotators. The inter-rater agreement turned out to be moderate for the simulated data (Krippendorff alpha 0.45: annotators agree on 55% of the samples) and poor for real data (K.a. 0.26: annotators agree on 41% of the samples). Thus, only simulated data could be used (with a caveat that the ground truth is agreed upon only half of the time).

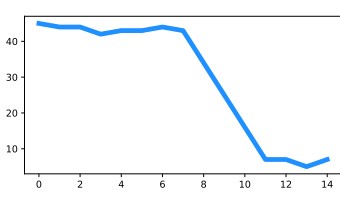
(a) **Human 1:** Peaks in the middle.
**Human 2:** Stays steady in the beginning.
**Human 3:** Peaks at the beginning.
**CATS:** **Variable shows a gradual decreasing trend throughout the entire duration.**
**Llama3.1**: The variable exhibits a general decrease in value over time, with some temporary increases and fluctuations around a downward trend.
**GPT-4o:** Variable exhibits a gradual decline with minor fluctuations early on.

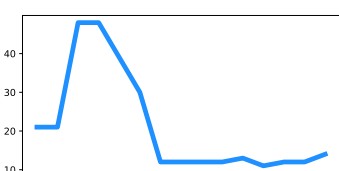
(b) **Human 1:** Peak at the beginning.
**Human 2:** Peaks in the beginning.
**Human 3:** Peaks in the beginning.
**CATS:** **Variable experiences a downward trajectory with an initial increase.**
**Llama3.1**: The variable exhibits a fluctuating trend with periods of increasing and decreasing values around a central level, resulting in a overall oscillating pattern.
**GPT-4o:** Variable rises sharply, peaks, and then gradually declines with fluctuations.

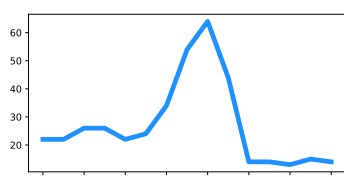
(c) **Human 1:** Peaks in the middle.
**Human 2:** Plot showing the hike in the mid time.
**Human 3:** Peak in the middle.
**CATS:** **Variable increases slowly before reaching a high level and decreases back to the end.**
**Llama3.1**: The variable exhibits an initial increase followed by a period of fluctuation and then a more pronounced upward trend, before eventually decreasing somewhat and stabilizing around a lower value.
**GPT-4o:** Variable shows initial growth, a sharp spike, and gradual decline.

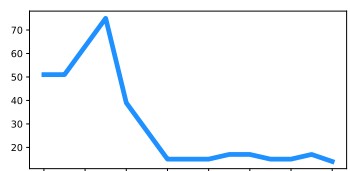
(d) **Human 1:** Sharp decline shortly after start.
**Human 2:** Sharp increase at the start.
**Human 3:** Peaks in the beginning.
**CATS:** **Variable gradually declines with minor fluctuations.**
**Llama3.1**: The variable exhibits fluctuations that result in periods of both increase and decrease, with a general tendency towards a mix of rising and falling values over time.
**GPT-4o:** Variable shows an initial rise, then a sharp drop, stabilizing later.

Figure 5: Examples of time series descriptions from the financial dataset. Summaries by CATS and GPT-4o give a complete picture, while human annotations are very short, repetitive, and often contradict each other

We tested CATS on the entire data in a zero-shot fashion, without retraining. In addition, since the data was in the public domain, we could also run a comparison with GPT-4o as a proprietary, but more powerful baseline than the ones approved for the confidential data. We also tested Llama3.1 as the stronger open-source candidate. The resulting TrendScore of CATS summaries was 0.59, close to Llama3.1 (0.60) and GPT-4o (0.62). Interpretation of these numbers requires a closer look at examples. As can be seen in Fig. 5, CATS-generated summaries are much more complete, specific and nuanced than the ground-truth annotations. They are also more advanced linguistically, with an average length of 9.8 words (cf. 4.9 in references) and Measure of Textual Lexical Diversity (McCarthy, 2005) of 40.52 (cf. 16.81 in references). This shows that CATS can generalize to new data and summarize time series in an accurate and detailed

|  | TRUCE | TSLM | GPT-4o | Llama3.1 | Human 2 | Human 3 | CATS |
|---|---|---|---|---|---|---|---|
| BertScore | 0.77 | 0.88 | 0.84±0.00 | 0.84±0.00 | 0.87 | 0.88 | 0.84±0.00 |
| TrendScore | NA | NA | 0.62±0.02 | 0.60±0.00 | 0.56 | 0.66 | 0.59±0.02 |
| Llama3.1-as-a-judge (F1) | NA | NA | 0.27±0.05 | 0.13±0.02 | 0.14 | 0.17 | 0.16±0.03 |
| Llama3.1 consistency | NA | NA | 0.72±0.01 | 0.75±0.02 | 0.65 | 0.66 | 0.75±0.01 |

Table 5: Results on TRUCE data. Scores for TRUCE as reported in Jhamtani & Berg-Kirkpatrick (2021), for TSLM — in Trabelsi et al. (2025). Annotators 2 and 3 are evaluated against annotator 1 as reference. BertScore is almost uniformly high, while LLM-as-a-judge F1 is low for all candidates, including its own output

way. It is also evident that GPT-4o is significantly more capable in time series description than the LLMs of previous generations, and even shows some advantage over CATS, which is not surprising considering the size of GPT-4o and its training data. What could appear surprising is the low TrendScore of both models — yet the examples shed some light: human annotations throughout the dataset show a clear pattern of describing one property of a sample (e.g. 'high' or 'low') and its location (e.g. 'in the beginning' or 'in the middle') in a very short and simple phrase, apparently following a provided example, with minimum variation. Such descriptions are incomplete even when all three annotators agree (which is only the case half of the time). This is problematic for the evaluation regardless of the metric used. In Tab. 5, we provide TrendScore, BertScore and Llama3.1-as-a-judge (binary F1) for comparison on the TRUCE data.

As discussed in Sec. 4.3.2, metrics relying on word embeddings are not suitable for evaluating time series summaries due to the similarity of embeddings of words describing opposite trends, such as 'rise' or 'fall'. This prevents existing metrics from reflecting misalignment between time series trends and their descriptions — see Fig. 2. The fairly high and almost uniformly distributed BertScore speaks more to the fact that all candidate sentences belong to the same topic, rather than to their individual semantic alignment with the input time series. As to TrendScore, it reflects the rather low inter-rater agreement of the ground truth annotations and is thus similarly low both for summaries written by the other humans and those generated by the models: given that the annotators agree only in 55% of samples, it is only fair that other annotators will be considered 'wrong' at least half of the time. Model scores are equally affected by this randomness in the ground truth.

By contrast, all scores of Llama3.1-as-a-judge are low (and are lowest for its own outputs), which is consistent with results reported in Sec. 5.1. Nonetheless, Fig. 5 clearly demonstrates that summaries generated by GPT-4o and CATS are more specific and complete than the human annotations in this dataset.

Overall, evaluation on the financial dataset results in several observations. First, both CATS and GPT-4o produce more *accurate and elaborate descriptions* than the crowdsourced annotations. Second, although accurate quantitative assessment is challenging in view of the above, visually, CATS performs almost on par with GPT-4o on time series description, while being *several orders of magnitude smaller* and being *trained only on 250 samples*, in contrast to the undisclosed but incomparably bigger training set of GPT-4o. Third, in addition to the advantages in terms of the size of the model and training data, crucially, it can be *deployed offline*, even on hardware with limited computational capacity, and *preserve data privacy*, thus *fulfilling the critical requirements of industrial applications and other domains*. Fourth, automatic evaluation of time series summarization is challenging not only technically, due to the versatility of natural language, but also 'epistemically', due to biases and limitations of the ground truth, which leads to the last observation. Namely, description of time series is a complex and challenging task not only for models, but also for humans.

# 6 Conclusion

In this work, we proposed a model and training method for efficient generation of accurate time series summaries, which can enable flexible multimodal user experience and communicate important trends in the development of a signal over time when visualization is problematic. This can provide insights into the underlying process dynamics, which is essential for making informed decisions quickly, particularly in complex industrial applications involving high risk to equipment, human safety and environment.

Our compact model, consisting of a time series encoder and a text decoder, allows efficient pretraining on unlabeled data, and the novel cross-modal autoencoding allows training an accurate model even on a very small dataset, without expensive labeling and prohibitively costly computations required by big models. The proposed cross-modal autoencoding may be further investigated as a basis for uncertainty estimation to boost user trust in model predictions.

As has been demonstrated in experiments on real industrial data with both automated evaluation and a comprehensive user study, descriptions generated by our model fulfill all the main requirements: they faithfully capture relevant properties of the time series and are concise and easy to read. Importantly, the model is fast, resource-efficient and can be deployed offline, preserving data privacy, which is a critical requirement in the industry, as well as in healthcare and other domains.

For further improvement in relevance and accuracy, additional pretraining techniques can be explored, such as cross-modal masking or contrastive approaches. Collecting more or better annotations (e.g., by native speakers, or even by more powerful LLMs using open data) would likely boost readability even further.

An extension of our approach to multivariate time series may be of interest, for instance, for financial use cases, such as comparing stocks. Technically, our approach is directly extensible to it by changing the number of channels in the encoder. The train set would need to be increased accordingly.

Our additional contribution is a dedicated metric, TrendScore, which quantifies how often the overall trend is captured. It is a very nuanced task, and TrendScore is an important first step, however, future work is needed for further improvement, for instance, it can be enhanced by allowing more fine-grained trend categorization, e.g., by assigning a sequence of trend classes and calculating the Levenshtein distance between the prediction and the reference. At the same time, as the evaluation on the financial dataset showed, due to the task complexity, even human annotations cannot be assumed to be reliable golden truth, therefore research of direct evaluations of texts given only time series (without labels) would be an interesting avenue, e.g. incorporating techniques like SAX as a starting point — or hierarchical classification of time series combined with partitioning. However, this is a vast and complex field. Nonetheless, to the best of our knowledge, the proposed metric is presently the only model-independent metric reflecting semantic accuracy of time series descriptions, aligned with human judgment.

A production pilot of CATS is planned as a basis for future work.

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
