# OpenReview forum: "CATS: Cross-Modal Autoencoding for Time Series Summarization"
_TMLR — Rejected by TMLR_

### Review · Reviewer_M4FX · 2025-07-30

**Summary Of Contributions:**

This work proposes an approach for producing textual descriptions from time series.  They employ an encoder-decoder architecture trained with a custom loss function, and multiple stages of pretraining and finetuning.  The model is applied to industrial and financial tasks.

## Strengths
I had not considered the task of time series description before.  It seems like a potentially important capability that merits further work, and thus I appreciate the high-level goal of this paper.

## Weaknesses
Overall, I think this paper has many significant weaknesses in its current form, in particular with respect to writing quality, motivation, and missing citations.

Below I list specific concerns:

- The abstract is very unclear with respect to what the specific research question being answered is.
- Given that time series description is not a widely known field, getting more background up front would be very useful.  For example, after reading the first sentence, I was left wondering what kind of descriptions we were considering, e.g., what granularity and with what goal. Some more concrete examples and use cases and intuition around the problem would be helpful before discussing methodology.
- NLG is not an abbreviation for natural text generation
- Abbrevation LM is not introduced
- The transition between the first and second paragraphs is jarring, from general use cases to industrial applications.  The second paragraph seems like the paper is going to narrow its focus to domain-specific industrial applications, but then it gets more general again.  The focus and goals of the paper should be tightened and made clear.
- The paper is lacking important citations in key places.  For example, I am left wondering if the entire second paragraph is simply the opinion of the authors, or supported by existing research.  The introduction has no citations at all after the first paragraph, and only one citation from after 2018.  By not establishing a clear existing research thread in this area, I have trouble identifying the audience for this paper.
- The opening phrases of the third and fourth paragraphs do not reflect what came before them: the third paragraph does not sum up the second, and it is not clear what “end” is being referred to in the fourth.
- While the method is motivated by a set of requirements for usefulness in this domain, these requirements are not connected to particular technical choices.  If the authors would like to argue that this approach is optimal according to different dimensions (e.g., speed, transfer), they should make clear what other options could be considered, and why specific choices are made.  E.g., how does cross-modal attention fit the goal of efficiency?  It would seem like some late-fusion approach might be more efficient.
- What does the arrow pointing to the plant operator mean in Figure 1?
- Section 3 lacks citations.  For example, where is the reference for autoencoding?
- I find the proposed TrendScore to be poorly motivated, and lacking evidence for its usefulness.  The problem of scoring open-ended generations is not new, and well-known solutions exist.  For example, you could use a small LLM as a judge.  More specifically, if there are only 6 classes of interest, why not just do classification?
- I find it hard to gain meaningful insight from the experiments, as I am unsure what the particular research question being investigated at this point is.  The focus on private data is not ideal either.

**Audience:**

No

**Audience Explanation:**

The authors fail to clearly establish an ongoing research story in this area, and actually make it seem like it is not an area of high interest.  While I think this is an interesting area, I do not think this paper can effectively introduce the TMLR audience to it.

**Broader Impact Concerns:**

None.

**Claims And Evidence:**

No

**Claims Explanation:**

No, I do not think that this paper offers strong evidence as to how the research community should approach building time series description models.  The methodological choices are not clearly explained, nor thoroughly tested.

**Requested Changes:**

Please see weaknesses listed above.  I think this paper would benefit from both a high-level sharpening of goals and motivations, and low-level editing and polish.

---

> ### Author Response · Authors · 2025-09-08
> **We appreciate the reviewer’s feedback and have carefully revised the manuscript (revisions in orange)**
>
> TS=time series; TSS=time series summarization; CMA= cross-modal autoencoding
>
> W1
>
> Thank you for asking! Given that TSS has remained under the radar of ML research, we felt it important to stress its industrial relevance on the one hand — and obstacles for development (lack of labeled data) and deployment (model size constraints and offline use due to privacy and cybersecurity) on the other. We investigate whether accurate, relevant and readable TS summaries can be generated with a small model trained on a small dataset, to fulfill real-world limitations. Abstract revised
>
> W2
>
> Adding more details and keeping the paper concise is always a trade-off. In addition to Fig.4 and 5, we added a textual example in the intro
>
> W3
>
> Thank you for catching the typo
>
> W4
>
> Thank you for paying attention
>
> W5
>
> Thank you for the feedback! In the 1st paragraph, we set the context for the problem by outlining the range of industries where it is relevant and reference previous work. Then we mention the limitations of both traditional and recent approaches.
> In the 2nd paragraph, we introduce the use case and background, which determine the requirements to TSS that we aim to fulfill.
> We revised the transitions to make them smoother
>
> W6
>
> We apologize for missing citations. Added
>
> W7
>
> We hope transitions now read smoother
>
> W8
>
> Thank you for pointing this out. Since TSS is a seq2seq problem, the choice of an encoder-decoder ar-chitecture with cross-attention was fairly straightforward: it has been widely used after it was first in-troduced [1]. Since TSS belongs to the field of multimodal translation (inferring one modality from an-other) rather than fusion (combining modalities to make a prediction) [2], using late fusion was not a good fit. S. rev. Sec.3.1.
>
> For the sake of completeness, in initial experiments we tried concatenating TS embeddings directly with text but abandoned that path in view of unsatisfactory results due to its poor fit to the small model size and low data regime (F1 0.107).
>
> Overall, rather than cross-attention, our main contribution in the context of data-efficient model train-ing is CMA (an alternative to which is basic training on text generation, which we showed to be insuffi-cient).
>
> [1] Vaswani, A. et al. (2017). Attention is all you need. Advances in neural information processing sys-tems, 30.
>
> [2] Baltrušaitis, T. et al. (2018). Multimodal machine learning: A survey and taxonomy. IEEE transactions on pattern analysis and machine intelligence, 41(2), 423-443.
>
> W9
>
> To differentiate between CMA vs subsequent normal training and inference, we indicate the flow of information with colors: blue for CMA (text embeddings are fed back into the encoder to reconstruct TS and compare to the original); green for normal training (with only TSS loss minimized) and inference (arrows from the plant and to operators highlight that this flow is the same after the model is de-ployed).
> Revised caption of Fig.1 to make it clearer
>
> W10
>
> We are a little confused about this. Autoencoding being a well-established method, attributing it to one source is problematic. In existing publications, most often it is not cited; sometimes [1] is mentioned, which cannot be correct, since [1] themselves (and some others) reference [2], although the term is used as early as in [3]. Thus, we are hesitant about citing any particular source.
>
> [1] Bengio, Y. et al. (2013). Representation learning: A review and new perspectives. IEEE transactions on pattern analysis and machine intelligence, 35(8), 1798-1828.
>
> [2] Hinton, G. E., & Salakhutdinov, R. R. (2006). Reducing the dimensionality of data with neural net-works. Science, 313(5786), 504-507.
>
> [3] Le Cun, Y., & Fogelman-Soulié, F. (1987). Modèles connexionnistes de l'ap-prentissage. Intellectica, 2(1), 114-143.
>
> W11
>
> S. rev. Sec.5.1 and 5.4.
>
> TrendScore uses six classes as an approximation, yet the task is to generate not only classes but rich, nuanced TSS required by operators
>
> W12
>
> Revised the abstract to make the RC clearer. We investigate if accurate, relevant and readable TS sum-maries can be generated with a small model trained on a small dataset, to fulfill the real-world limita-tions.
> The three quality aspects (accurate, relevant and readable) are evaluated in a user study (Sec.5.3). Accu-racy is additionally quantified in Sec.5.1.
> Competitiveness of our approach despite the small model size is shown through comparison with x35 larger baselines, and the efficiency of the proposed CMA is evaluated in an ablation study (Sec.5.2).
>
> Regarding private data, we would be happy to use open data, and we do test our model zero-shot on the only open dataset (and we discuss its limitations in Sec.5.4). Had curated TSS datasets existed, it would have spared us the effort of collecting our own. Unfortunately, this is not the case, and this is what motivates our work on efficient pretraining on limited data

---

> > ### Comment · Reviewer_M4FX · 2025-09-09
> > **reviewer acknowledgement**
> >
> > I thank the authors for their detailed rebuttal and revision.  My main concerns about the paper remain largely intact.
> >
> > I find the authors do not present a sharp machine learning research question that is of clear interest to the TMLR audience.  It may be the case that this paper is best suited for an IEOR (or other industry-focused) venue.  Also, wherein I can discern some goals around industrial criteria for these systems, methodological decisions are not clearly linked to these goals.

---

> > > ### Author Response · Authors · 2025-09-11
> > > **We thank the reviewer for the thoughtful comments and for the opportunity to clarify the scientific contribution of our work**
> > >
> > > In response to your comment on a sharp machine learning research question that is of clear interest to the TMLR audience:
> > >
> > > We tackle a fundamentally underexplored ML problem of time series summarization (TSS), which, despite decades of effort on rule‑based systems and more recent data-driven attempts, remains largely unsolved due to its complexity and the lack of labeled data (Sec.2). Consequently, there is a clear gap: how can we learn a compact model that jointly reasons over temporal dynamics and produces concise, accurate, and readable text? This question lies at the intersection of representation learning, cross‑modal generation, and evaluation methodology.
> > >
> > > Since the value of TSS is in giving the user an idea of the overall pattern in a time series for quick triage and decision support without seeing a plot, we argue that:
> > >
> > > 1. The training technique must encourage alignment of time series and text. We consider basic training on text generation using categorical cross-entropy loss insufficient since embeddings of words critical for faithful TSS may be close even if their semantics are opposite and since a difference in one word would not prevent loss minimization on the whole sequence (s. example in Sec.3.2). To bridge this gap, we propose a novel technique of training the model to generate descriptions that enable reconstruction of the specific input time series.
> > >
> > > 2. The evaluation of TSS can only make sense considering alignment of time series and text, which manifests itself in trend words, and offer an evaluation method grounded in such words (s. Listing 1, Sec.4.3.2).
> > >
> > > Thus, the work promotes a deeper understanding of a real-world problem, as well as approaches to solving and evaluating it. It reveals the challenges that have not been acknowledged or addressed in existing research and offers methods to tackle them.
> > >
> > > Our manuscript therefore tackles a newly defined problem setting: faithful time‑series captioning under realistic industrial constraints, which is of interest to the TMLR community because it:
> > > 1. Expands the scope of cross‑modal generation beyond vision‑language tasks, introducing unique challenges.
> > > 2. Highlights a mismatch between conventional NLG evaluation and cross‑modal fidelity, prompting the community to reconsider metric design for multimodal generation.
> > > 3. Offers a principled learning method — cross‑modal autoencoding (CMA) — that can be adapted to other domains, thereby opening a new research direction.
> > >
> > >
> > >
> > > Regarding the methodological decisions being linked to industrial criteria:
> > >
> > > In this work we address the following critical requirements to time series summarization: faithfulness to input, readability, prediction speed and efficiency, as well as independent offline deployment preserving data privacy.
> > >
> > > Faithfulness to input is the core motivation of the proposed CMA method and it is also the criterion addressed by the proposed evaluation method.
> > >
> > > Readability is an implied requirement, since unintelligible descriptions would be of no value. It is accounted for by using a pretrained text decoder (GPT-2), which is sufficiently powerful for the task without being computationally expensive. We run a user study to explicitly evaluate both faithfulness (as a combination of accuracy and relevance) and readability.
> > >
> > > Finally, practical considerations including prediction speed, efficiency and independent offline deployment motivate the choice of compact encoder and decoder, as opposed to large LMs: while 7B models are infeasible in real applications, small-scale 0.24B models are realistic.
> > >
> > > Since model performance heavily depends on its size and the training data, and in real applications both are severely limited, we propose a training method which aims to compensate for these constraints by maximizing cross-modal representation learning even with a small model and little data.

---

### Review · Reviewer_N6id · 2025-08-12

**Summary Of Contributions:**

**Summary**

The authors study the problem of describing time-series using natural language. Specifically, they target an industrial setting where local LLMs have to be used (due to privacy reasons), and where inference time is critical. They propose a method called Cross-modal Autoencoding for Time series Summarization (CATS), which combines a PatchTST time-series encoder with a GPT-2 text decoder via cross-attention. They then propose a cross-modal autoencoding loss which they use to train this model on paired data. The authors evaluate their method on two datasets, finding that they outperform the baselines on both a metric they propose and a human evaluation.


**Strengths**
1. The motivation of the paper is strongly driven by real-world industry needs (e.g. low inference time, local models), and this is reflected in the evaluation.

2. The paper is generally clear and easy to follow.

3. The authors do a good job of explaining why existing methods like BLEU and BERTScore would not work in this setting (Table 1 and Figure 2), and thus why a new metric is necessary.


**Weaknesses**
1. There are significant weaknesses in the evaluation. First, the authors propose both a new _dataset_ and a new _evaluation metric_. As the TrendScore metric is simply based on matching keywords, it is easy to _bias_ this metric towards the keywords in the dataset in which the authors have collected. This is especially important since the large proportion of baselines are zero-shot prompted LLMs, which do not see any examples of how the answer should be phrased. As such, CATS would have an artificially inflated TrendScore as it can output the same keywords it sees during training. For example, in Figure 4 (d), we see Gemma providing an actual value (418.7) for the range of the time-series. These values seem very useful for an operator looking at an instrument. However, it is not rewarded for doing so because it is not measured in TrendScore. Though the authors do conduct a human evaluation, the inter-rater agreement is rather poor (0.337).

2. The method has only been evaluated in the univariate setting, which greatly limits is utility.

3. The set of possible descriptions assigned to a time series ("increasing; decreasing; stable; increasing and then decreasing; decreasing and then increasing; and fluctuating") is extremely simplistic, and I do not believe it covers all use cases across real-world time series. What about examples like "the time-series became more/less noisy over time", or "the frequency of the oscillation increased/decreased"?

4. The zero-shot prompting LLM baselines used in the paper are rather weak. The authors motivate the problem as requiring local LLMs for privacy reasons, where inference time is important, and so only use 7B LLMs. This is reasonable, but the authors should use more recent LLMs with better capabilities such as Llama-3.1-8B-Instruct or Qwen2.5-7B-Instruct, instead of only Mistral 7B and Llama 2, both of which are 2 years old. The authors should also clarify whether they are using the instruction tuned versions of these models.

5. The authors should add additional baselines which go beyond the zero-shot LLM prompting setting. This would allow them to demonstrate the empirical advantage of their cross-modal loss over more competitive baselines. One such baseline is [1] which projects time series to the LLM embedding space. In Section 2, the authors also mention a prior work using "vision as bridge (plotting time series and using vision-language models)". Is there a reason why using CLIP with this method was not tried as a baseline?



[1] Towards Time-Series Reasoning with LLMs. arXiv:2409.11376.

**Audience:**

Yes

**Audience Explanation:**

Yes, the paper would be interesting to people working in time-series who are interested in integration with LLMs, especially those interested in more industrial applications.

**Broader Impact Concerns:**

No broader impact statement is necessary.

**Claims And Evidence:**

No

**Claims Explanation:**

There are several issues with the evaluation (W1), utility of the method (W2-3) and baselines (W4-5) that should be resolved.

**Requested Changes:**

Critical:

1. The authors should add stronger baselines, both stronger LLMs for prompting (Weakness 4) and baselines from prior work (Weakness 5).

2. The authors should add an alternative to TrendScore, as there is potential for bias/gaming (Weakness 1), and the human evaluation is not convincing due to rather mixed results (Table 3), and low inter-rater agreement. One option is to use a frontier LLM like o3 as a judge, passing in the ground truth and the model output (along with potentially the time series), and asking it to rate the model output.

3. I would like to see the method evaluated in more complex scenarios, either with more complex descriptions (Weakness 3), or in the multivariate setting (Weakness 2).

4. Please add confidence intervals to all results. This might be computed e.g. through a binomial CI.

5. Please comment on the connection to [2], which proposes both a dataset for time-series QA and a method. It seems like their method can be directly integrated as a baseline.

6. There are some critical experimental details missing which should be added, including but not limited to: what value of $\alpha$ was used and how was it chosen? What was the prompt given to the local LLMs and gpt-4o? How exactly is TrendScore computed (e.g. stemming, negation handling, presence of multiple conflicting keywords, etc)?

Minor:

7. Please provide the list of 148 keywords used in TrendScore in an appendix.

8. Please comment on whether you plan on publicly releasing the code and the dataset you collected.

[2] ITFormer: Bridging Time Series and Natural Language for Multi-Modal QA with Large-Scale Multitask Dataset. ICML 2025.

---

> ### Author Response · Authors · 2025-09-08
> **We thank the reviewer and provide detailed answers (manuscript revisions in purple)**
>
> TS=time series; TSS=TS summarization; TSD=TS description; CMA= cross-modal autoencoding; CLF= classification; TrSc=TrendScore
>
> W1
>
> We fully understand the concern regarding keywords and manually verified classes in test set (Sec.4.3.2).
> The 148 keywords, collected within the scope of a work not yet published, cover all trend words used by CATS and LLMs (and more).
> To avoid any bias, we conducted a user study for a nuanced evaluation.
> On TRUCE data, both CATS and GPT-4o are tested zero-shot and TrSc is not biased towards CATS.
>
> As to numeric values in TSDs: we prompted models to *avoid* references to exact times and values, since capturing the main patterns in TS brings the most value to operators, while statistics like min, max etc. can be calculated deterministically. In addition, evaluating accuracy and relevance of numeric values is a different task: estimating 1) if a variable truly takes value V at time T, 2) how relevant it is overall and if a text correctly describes what property of TS V conveys. Cf.Fig.4(a): Mistral mentions concrete numbers, yet the overall summary is confusing.
>
> The moderate inter-rater agreement reflects the task complexity and may also be due to the Likert scale.
>
> TSS is a very complex problem, which does not get enough attention in the ML community due to the lack of benchmarks. That's why we offer TrSc as a first attempt at a fair evaluation of TSS, which could improve further in future work
>
> W2
>
> Thank you for pointing this out. We, indeed, focus on univariate TS, which is useful in the industry: e.g. ensuring pressure in a tank does not exceed a safety threshold, or optimizing ore recovery KPI. If more variables are of interest, they can be described individually and combined with rules. Multivariate TSS may be relevant in finance, e.g. comparing stocks. CATS is directly extendible by changing the number of encoder channels. The train set would need to be increased accordingly and evaluation would require extensions (syntactic parsing etc.)
>
> W3
>
> We agree that 6 classes cannot be exhaustive and mention in Conclusion that TrSc is important as a first step and discuss future directions. That being said, TS CLF, change point detection and other tasks required for evaluation are all vast fields of research in their own right and practically, a metric capturing the overall TSD quality is valuable. Cf. BLEU and other established metrics, which heavily depend on references, their length, synonyms, grammar etc. and may be completely off at the level of individual sentences yet are typically useful on a larger scale.
> Overall, we believe TrSc offers a more discriminative assessment than e.g. BertScore and proposed extensions can further enhance its ability to capture nuanced TS characteristics
>
> W4
>
> Indeed, we are obliged to use the approved open-source LLMs on‑premise. The approval process takes time and we used all available models. Llama3.1-instruct is now also approved: added to Sec.5.1 and 5.4.
>
> For TRUCE, we compare CATS with GPT-4o (Sec.5.4). We find that TSDs by both CATS and GPT-4o are more relevant than the ground truth and that CATS is competitive with GPT-4o despite the small size and training on only 250 samples.
>
> Yes, we use instruction-tuned LLMs
>
> W5
>
> Thank you for referencing. Added to Sec.2. Unfortunately, [1] lacks details and code for comparison. Yet their findings are in line with ours: 'encoding TS directly is more efficient than converting it to text or an image'[1]. Both approaches ([1] and ours) are similar in that they combine a TS encoder with an LM in-stead of passing TS as a string. Yet, they differ in scale and focus: Mistral in [1] enables long detailed TSDs, while CATS, with GPT2, focuses on the main properties of TS while being compact and efficient and thus suitable for production hardware.
>
> Initially, we also tried concatenating TS and text embeddings, but abandoned that path in view of un-satisfactory results, due to the small model size and low data regime (F1 score 0.107).
>
> Note that TSS in [1] is not evaluated: TSDs are used in CoT CLF, although the example (Appendix G) shows that although both candidates are long and overall valid, from the text alone, a user may picture a very different TS.
>
> S. Sec.4.2
>
> RC1
>
> S. W4 and W5.
> S. also scores of TRUCE and TSLM (Tab.5)
>
> RC2
>
> S. W1.
> S. rev. Sec.5.1 and 5.4
>
> RC3
>
> S. W2 and W3
>
> RC4
>
> S. Tab.2 and 5
>
> RC5
>
> Thank you for sharing. Data and code are provided, but are focused on QA, not TSS, and data has no TSDs. Yet [2] provides interesting results: 1) multimodal GPT4o and Gemini scores are very low (s. W5); 2) BLEU/ROUGE are biased towards the data; F1 is calculated on multiple-choice questions, which are dataset-specific and unreliable: in our tests with LLM-as-a-judge, LLM's decisions are very inconsistent
>
> RC6
>
> S. Sec.3.3, 4.2, 4.3.2
>
> RC7
>
> To be released with the yet unpublished work
>
> RC8
>
> While code and data are proprietary, part of the data stems from an open-source simulator, which we can share

---

> > ### Comment · Reviewer_N6id · 2025-09-09
> >
> > Thank you for the response and the new experiments! This resolves some of my concerns, though some of the critical concerns remain:
> > 1. RC3 (evaluation in more complex and realistic settings), and has also been echoed in W1 by Reviewer rxzb.
> >
> > 2. Issues with TrendScore (W1 and RC7). This is compounded by the fact that the authors are not able to release the full details of how TrendScore is calculated. This is concerning as TrendScore is the primary evaluation metric used in the work, and we are unable to fully validate it in this review process. If TrendScore is to be published in another work, that also reduces the novelty of this work (Strength 3).
> >
> > In addition, the authors have stated that they do not plan on releasing the code and part of the data. I find this quite concerning, as it limits reproducibility and potential interest of the work to the TMLR audience.

---

> > > ### Author Response · Authors · 2025-09-10
> > > **Thank you for the quick feedback! TrendScore code added**
> > >
> > > 1. The time series we used is real production data from customer whose request inspired this work, and it is very representative of the data one deals with in production. Nonetheless, we would also be interested in evaluating CATS on different time series, although the challenge remains that only one open time series description dataset could be found, and with much simpler descriptions.
> > >
> > > That being said, the component responsible for time series representation learning in CATS is the time series encoder. We used PatchTST, which has shown robust results across a variety of datasets [Nie et al., 2022]. Hence, we hypothesize that our approach would generalize well to different time series.
> > >
> > > 2. We thought the question was high-level. We certainly can share the code for TrendScore calculation: we added Listing 1 to Sec. 4.3.2. Also, to clarify: only the full keyword lists are part of the other work, not the TrendScore.

---

### Review · Reviewer_rxzb · 2025-08-28

**Summary Of Contributions:**

This paper proposes Cross-Modal Autoencoding for Time Series Summarization (CATS), a learning model tailored to generate textual summarization in the context of trend-like data relevant to time series.  Specific to time series, the authors propose a loss function that combines the cross-entropy of the text with the mean squared error of the time series prediction, thereby balancing the quality of text generation with accuracy in relation to the time series. They propose a new metric TrendScore, that measures the trends occurring in the underlying times to be summarized, and the efficacy of their approach in an industrial and financial application.


## Strengths
- **[S1]**: Novelty and motivation for loss function and TrendScore.  Overall, one major strength of the paper is in the proposed loss function and evaluation metrics used for the analysis.  While they do exhibit some limitations, as I'll discuss later, they are well-motivated as a means to capture the trends relevant in the summarization of time series that classical metrics, e.g., standard BELU, would not prioritize.
- **[S2]**: Efficiency.  Overall, the proposed method is relatively lightweight, i.e., it utilizes a time series encoder (PatchTST) and a text decoder (GPT-2), making it more applicable to use cases with limited (but still non-negligible) computational resources.
- **[S3]**: Efficiency.  Overall, the proposed method is relatively lightweight, i.e., it utilizes a time series encoder (PatchTST) and a text decoder (GPT-2), making it more applicable to use cases with limited (but still non-negligible) computational resources.


## Weaknesses
- **[W1]**: Usefulness.  Overall, this is the biggest concern I have with the current version of the paper.  In the context of summarizing a time series through text, a question that I have is whether it is even useful.  For example, looking at the test cases for both datasets (Figures 4 & 5), the graphs convey much more information and take less time to interpret than reading the text summarization, prompting me to be skeptical of whether we would even want to use machine learning for this application in the first place.  Considering this in an industrial setting (Figure 1), where a plant operator is present to inform field operators based on the time-series data, I see this even less helpful, as their expertise and domain knowledge would likely be of much higher value than any of the summarization produced.   Perhaps one way to overcome this would be to summarize more complex time series examples quickly; however, as it stands, I don't see the current examples as demonstrating a practical use case for the underlying problem.
- **[W2]**: TrendScore.  Overall, I believe the concept of a metric like TrendScore is beneficial in capturing time-series relevant text summarization characterization beyond standard natural language metrics.  However, it is limited in only capturing a few types of variability, which ultimately makes it only applicable to measure very examples.  For example, in an industrial context, there may be a significant difference in downstream decision-making based on the rate of change, e.g., whether it increases slowly or quickly, but this would not be captured in TrendScore.  Broadly speaking, I think trying to capture the nuances of time-series data is actually quite challenging.  For example, referencing Figure 5, I argue that none of the Human- or CATS-generated text accurately captures the underlying time series, while GPT-4o does a reasonable job.  Given that they all report a very similar TrendScore, I am skeptical about the actual usefulness of this metric.
- **[W3]**: Dataset & Human Scoring: Overall, I have limited confidence in the dataset and evaluation setup. Human-generated summaries of time series are prone to inaccuracies (see Figure 5 for example), and the industrial dataset is very small (148 samples across six classes). Coupled with the relatively low inter-rater agreement reported, it is difficult to draw strong conclusions about the actual effectiveness of CATS.
- **[W4]**: The paper argues that CATS is practical because it uses GPT-2 as a lightweight decoder. While this may hold for offline servers or operator workstations, in many industrial contexts (e.g., controllers or edge devices), even GPT-2 would be unrealistic. A more nuanced discussion of where the model can and cannot be deployed would make the contribution more convincing.

**Audience:**

Yes

**Audience Explanation:**

The topic is relevant to the machine learning community, albeit somewhat niche, given its focus on summarizing time series.  That said, I believe it to be a good fit for TMLR.

**Broader Impact Concerns:**

I have no broader impact concerns.

**Claims And Evidence:**

Yes

**Claims Explanation:**

Overall, the claims are reasonable.  However, I have low confidence in the data (see weaknesses), so I'd argue they are not that convincing.

**Requested Changes:**

Overall, I'd need to see a much more rigorous evaluation with concrete examples to have confidence in the novel metric (TrendScore) and the summarization performed by the proposed model.  Moreover, I'd need more convincing evidence that time-series summarization is useful, either through the capability to summarize much more complex series or well-motivated industrial examples.

---

> ### Author Response · Authors · 2025-09-08
> **We appreciate the reviewer’s feedback and offer clarifications below**
>
> TS=time series; TSD=TS description
>
> W1
>
> We agree that a picture is worth 100 words and a plot is more informative than TSD — just like image captioning or text summarization. Nonetheless, in many cases it is useful: e.g., control room operators often monitor an entire plant with an overwhelming number of processes on multiple screens (from 5 to 50). In case of abnormalities, the system triggers an alarm or warning to call their attention. Howev-er, many alarms are false positives [Hollender, M. Collaborative process automation systems. ISA, 2010.]. In such cases, a 'non-invasive' short summary of a problematic signal as a text or text-to-speech message can help the operator quickly estimate if an alarm needs closer attention. In addition, modern systems may run ML models, and under the new EU AI Act their developers may be obliged to explain model predictions, e.g. indicating which signal or pattern contributed most. In this case, a short TSD can help the operator make an informed decision. And for field operators, who don't have any screen at all, a TSD may be the only way to be informed about a given signal (control room operators would not be describing process variables for them, since their own work requires their full attention — we apologize if we haven't made the distinction clear). While in-depth study of complex signal interactions indeed is most convenient through plots, it is short text summarizing the main patterns in TS that is often best suited for real-time communication in production environment.
> In addition, in Sec.2 we cite several complex rule-based systems for TSD (and many more exist). They were developed despite the high cost, driven by industry demand. And compact offline models are re-quired wherever the use of proprietary LLMs is prohibited due to privacy policies, also in other domains, such as healthcare or finance
>
> W2
>
> S. Sec.5.1 and 5.4.
>
> Indeed, the endless variety of TS cannot be limited to six classes, and we mention in Conclusion that TrendScore is important as a first step and discuss future directions, including combining sequences of classes with edit distance, or partitioning TS to estimate the relative size of a trend within a window, etc. However, TS classification, change point detection and other tasks required for a robust evaluation are all vast fields of research in their own right, and for practical purposes, a metric capturing the overall quality of a TSD is valuable (and concise TSDs capturing the main trends in a signal are most useful in practice — in fact, we prompt all models to describe only the main properties of a TS). Cf. BLEU, ROUGE and other well-established NLG metrics, which heavily depend on references, their length, synonyms, grammar, etc. and may be completely off at the level of individual sentences yet are typically useful in large-scale evaluations.
>
> W.r.t. TrendScore distribution in Tab.5, it is due to the TRUCE dataset quality (both simplistic TSDs and low annotator agreement), as we explain in Sec. 5.4. Like all other metrics (BLEU, BertScore, etc.), TrendScore relies on references and thus depends on their quality. Nonetheless, TrendScore offers a more discriminative assessment than generic metrics. Cf.: BertScore is almost uniformly high for all models on both datasets (and, in fact, biased in favor of CATS), yet all it tells us is that predictions and references pertain to the same topic without differentiating among TS properties (s. Sec.4.3.1)
>
> W3
>
> We would like to clarify the numbers:
>
> 1) The data we collected contains 416 samples. Each test sample is evaluated by 5 users; the agreement of 0.337 is considered moderate and reflects the task complexity. It may also be due to the choice of Likert scale.
>
> 2) 148 is the number of keywords used in TrendScore (used both independently, in combination with each other and in conjunction with location - 'beginning', 'middle', etc. — we added a comment in Sec.4.3.2).
>
> 3) Fig.5 shows samples from the financial data that we used in addition to ours. We fully share your opinion of the subpar quality of the crowdsourced annotations in that dataset. Unfortunately, depend-ence on the ground truth quality is the problem with all benchmarks. We show prediction examples (Fig.4 and 5) to better illustrate caption quality
>
> W4
>
> While field devices and controllers are, indeed, not meant for running neural nets, at supervisory level, plant control systems can have sufficient computing power to run ML models, and lower-level devices in the network can request data from them. It is worth noting that due to high development costs and conservative regulations, at present, few systems have ML models running in production, but as more models and pilot deployments are developed, industries also gradually transition to more powerful hardware

---

> > ### Comment · Reviewer_rxzb · 2025-09-22
> > **Response**
> >
> > Thank you for the clarifications and discussion.  I still have some concerns that I will discuss below.
> > - [W1]: Based on the explanation, it seems even less clear as to why natural language would be preferred over a signal indicating the class of the TDS.  In my view, this would provide as much information, unbloated by text, so perhaps considering this as a pure multi-class classification would be more appropriate.  With more complex time series and scoring metrics, my perspective would change.  However, as it is, I am still not convinced of the practical use for TSD specifically.
> > - [W2]: I mostly agree with the authors here in that there will be limitations in the first metric to evaluate TSD.  However, I am still not confident in TrendScore for the reasons discussed in my original review.
> > - [W3]: Thank you for the clarification on TS and the additional figure.  I share the authors' opinion that there is a limited amount of quality data.  This is primarily the reason for my low confidence in any results.
> >
> > Overall, I think the paper is interesting and has the potential for use cases.  However, my confidence in the data quality and evaluation is low.

---

> > > ### Author Response · Authors · 2025-09-23
> > > **We thank the reviewer for elaborating on their concerns and provide our responses below**
> > >
> > > W1
> > > We fully share your view that different use cases may require various degrees of granularity in a TS representation – from a class label to a more or less detailed description, to cases where only a plot could be of use. In our experience with various industries, use cases and user categories, the full spec-trum has its applications. We agree that a simplistic TSD like ‘Variable peaks in the middle’ may, indeed, be mostly covered by two sets of classes (one for the trend, like ‘increase’, the other for its location, like ‘end’), although even such basic TSD find their use cases in the financial domain [Jhamtani & Berg-Kirkpatrick, 2021]. At the other extreme, a highly detailed text may be harder to digest than a plot of raw TS (e.g. ‘The time-series data shows a general upward trend with fluctuations over a period of approximately 300 units. Initially, the values start around 1500 and gradually increase, reaching a peak of approximately 3500 around the 100th unit. Following this peak, there is a significant decline, with values dropping to around 1500. After this decline, the series exhibits a series of smaller fluctuations, oscillating between 1500 and 2500. Towards the end, there is a slight upward trend, with values hover-ing around 2000...’ [Chow et al., 2024]). By contrast, a concise but nuanced description like ‘Variable decreases slightly and stabilizes for a short time, then rises again’ (Sec.3.2) is both more specific than just a class (indicating how strong the trends are, how long they last, and in what order they appear) and sufficiently terse to be easily grasped, which is important under time pressure and stress in pro-duction environment (according to classical psychological studies, memory span averages to 15 words [Brener, R. (1940). An experimental investigation of memory span. Journal of Experimental Psychology]). Due to the balance of detail and conciseness, TSD we focus on are a perfect fit in cases when plotting is complicated (because adapting existing control system interfaces is a long and complicated process due to all the bureaucratic, legal and cybersecurity procedures) or impossible (when user does not have access to a graphical display). Thus, in the industrial domain, such summaries can ensure relevant deci-sion support both to control room operators and to field operators.
> > >
> > > W2
> > > Indeed, like you mentioned, TrendScore would capture the main properties of TSD and not nuances like the rate of change – just like any automated metric, it is an approximation (all models are wrong, but some are useful). We argue that capturing the overall correctness in a description of trends brings much more information than a generic n-gram or embedding-based metric. We already mentioned po-tential directions for enhancement, such as extending the score to trend sequences. Other ideas are possible, such as using a syntactic parser to extract adjectives and adverbs in phrases describing trends (e.g. ‘decreasing rapidly’), however, such a granular approach would be quite brittle for a num-ber of reasons: 1) any automated metric is constrained by the ground truth, which may not include such adverbs, or may instead use a different synonym to convey the same semantics (e.g. ‘plummeted’); 2) ground truth may have spelling errors which would corrupt the parsing and lead to wrong results; 3) one would need to decide on what weight to assign to such modifiers compared to the actual trend words, scaled either per phrase or per sentence. Therefore, although engineering an all-encompassing metric is an exciting exercise, it is prone to many pitfalls while only adding incremental value to captur-ing the overall trend. For this reason, we opted for an approach that is both more robust and focused on the most salient TSD properties – and significantly more informative than existing metrics.
> > >
> > > W3
> > > Exactly! Quality data is hard to come by. This is the reason why, in addition to the automatic evaluation, we ran a human study with qualified users including 22 Master’s students and 23 researchers and engineers (all with PhD degrees and years of experience in the field) – ensuring much more competence and motivation than if we had randomly selected Mechanical Turk annotators. Cf. most human studies in NLP involve between 1 and 5 annotators and a median of 100 samples [Van der Lee, C. et al. (2021). Human evaluation of automatically generated text: Current trends and best practice guidelines. Computer Speech & Language; Tam, T. Y. C. et al. (2024). A framework for human evaluation of large language models in healthcare derived from literature review. NPJ digital medicine]. In our case, each of the 100 TS samples (with several candidate descriptions per sample) was evaluated by 5 users, with 50 users overall. We hope that the reliability of the user study design compensates for the modest data size. We monitor publications for any new datasets and will be happy to test if any quality data becomes available!

---

### Comment · Action_Editor_xpEw · 2025-08-29
**Thank you reviewers!**

As we kick off the author-reviewer discussion phase for this paper, I wanted to thank the reviewers for their on-time delivery of well written reviews. You have given the authors good feedback that, as addressed, should help strengthen the paper.

I ask that you remain engaged throughout this phase of evaluating the paper. We owe that attention and respect to the authors as we are helping ascertain (and improve!) the value of the work that has been submitted for review. Stay tuned for author responses.

---

### Comment · Action_Editor_xpEw · 2025-09-12
**Please continue Author-Reviewer Discussions!**

Hi everyone,

I'm very pleased to see the productive discussions that have been had so far between the authors and reviewers. Unfortunately, this has not been a process that all reviewers have engaged in.

Note that even though the reviewers have been prompted to submit their recommendations, *I will not accept any recommendation from a reviewer who has not engaged in a good faith discussion with the authors*. It is our responsibility in the peer review process to recommend ways to improve the work and to engage in discussions to clarify points of misunderstanding. The authors have been diligent in updating the paper in response to reviewer comments and have continually sought opportunities to address reviewer concerns. We can offer their work the respect it deserves.

Thanks,
AE

---

> ### Author Response · Authors · 2025-09-12
> **Thank you for encouraging further improvement of our work!**
>
> We appreciate all the interesting questions and constructive criticism of the reviewers and are happy to address their concerns and further improve the manuscript.
> And we thank the action editor for organizing and supporting the discussion.

---

### Decision · Action_Editor_xpEw · 2025-10-03

**Recommendation:** Reject

**Audience:**

No

**Audience Explanation:**

Two of the three reviewers rated that this paper, if properly evaluated, would be of interest to the ML community as the problem the proposed metric and methods are designed to address are interesting and relevant. However, these reviewers were concerned about the lack of transparency about data, code and other analytical artifacts. For these reasons, they did not feel overly confident that the TMLR audience would be adequately interested in the work presented by the paper in light of the other limitations expressed in their reviews.

The third reviewer felt that this work and the insights generated therein may have a better audience in more industry focused venues such as IEOR.

**Claims And Evidence:**

No

**Claims Explanation:**

The only evaluations presented in the paper feature univariate time-series and where the set of possible descriptions have been assessed to be too simple.

Additionally, there were broad concerns about the reliability of the proposed TrendScore metric. Primary concerns are that the metric does not adequately account for potential biases nor is sufficiently applicable outside the narrow use case presented in this paper.